# A Tale of Two Identities: The Value, Attitude, and Behavior of Adult Children towards Family Tourism Experiences

Anan Hu, Houqi Li and Jinyuan Pang *

Department of Tourism, Fudan University, Shanghai 200433, China; huanan@fudan.edu.cn (A.H.);
20307080017@fudan.edu.cn (H.L.)
* Correspondence: 17210500032@fudan.edu.cn; Tel.: +86-150-0010-0694

**Abstract:** Adult children accompanying their parents on trips is a particular form of family tourism. During family travel, adult children assume two roles: as tourists, they pursue personal hedonic experiences, while as children, they also bear the responsibility of showing filial piety towards their parents. These two roles entail inherent contradictions. How this conflict influences the formation of family tourism values between adult children and their parents, and ultimately impacts adult children's intention to accompany their parents on future trips (re-travel intention), requires further in-depth analysis. Based on the perspective of tourist-to-tourist interaction and role conflict theory, this study applied a "value–attitude–behavior" model to empirically analyze the relationship between the interactions of adult children and parents during the tourism and their re-travel intention. An empirical investigation was conducted with 566 adult children from Jiangsu, Zhejiang, and Shanghai. The result of the structural equation modeling (SEM) analysis indicates that both positive interactions and role conflict have significant impacts on the formation of family tourism values, and these impacts are moderated by self-efficacy. Furthermore, these two variables influence attitude through the values of filial piety and family connection, which, in turn, affect the re-travel intentions of adult children. This study confirmed that the higher the self-efficacy of adult children, the better effect the positive interaction has and the less impact the role conflict has, which ultimately affects adult children's re-travel intention. At the theoretical level, this study reveals the process of the formation of "adult children–parents" family tourism values, and provides practical insights for family tourism marketing.

**Keywords:** family tourism; adult children; tourist-to-tourist interaction; role conflict; re-travel intention

## 1. Introduction

Family tourism is a type of tourism characterized by the participation of one or both parents together with their children. Although family tourism is a common form of tourism, it has long received relatively limited attention in the field of tourism research. However, with the continuous development of the family tourism market segment, family tourism has attracted more and more attention from tourism researchers and practitioners [1–3], and its capacity to enhance personal life satisfaction and foster stronger family bonds has gradually been recognized [4,5]. This trend is particularly pronounced in East Asian countries influenced by Confucian culture. This culture takes parents' being affectionate towards their children and children's expressing filial piety toward their parents as a fundamental principle when handling intergenerational relationships. Therefore, regions influenced by Confucian culture still place significant emphasis on family development, which has propelled an increasing popularity of family tourism [6,7]. However, the emphasis on filial piety inherent in this culture also results in local adult children (children who are 18 and above) developing value orientations that may differ from those of adult children in other regions when accompanying their parents on trips. In China, the birthplace of Confucian culture, traveling with parents has become a significant means for Chinese children to

provide spiritual and emotional support to their parents [8,9] and express their filial piety (an enforcement of obedience among the younger generation in an authoritative hierarchy built upon multi-generations) [10].

Based on the perspective of adult children, this study focuses on a distinct form of family tourism known as "adult children–parents" family tourism, which involves adult children traveling alongside their parents in contrast to the conventional parent–child tourism, where parents undertake trips with their underage children. Current studies have named this form of family tourism differently and a unified understanding of which family members are involved in this type of tourism is yet to be established. Drawing from previous studies' nomenclatures and conceptual definitions, the term "adult children accompanying parents on traveling" is abbreviated as "adult children–parents" family tourism for this study. This kind of tourism refers to family tourism marked by the full participation of both adult children and their parents, excluding instances where adult children sponsor travel activities instead of accompanying their parents. Notably, the term "parents" solely pertains to the parents of the adult children and does not contain their parents-in-law from this context, and the participants in this type of tourism must include, but are not limited to, adult children and their parents.

When traveling with parents, adult children are confronted with a series of conflicting choices that are not common in "ordinary" tourism activities. As travelers, adult children naturally seek personal gratification, such as entertainment or leisure seeking [11]. Nevertheless, for Chinese adult children, the pursuit of this "individualistic pleasure" may not be the foremost incentive during such journeys [12]. In contrast to Western societies, where individualism is highlighted, Eastern culture places significant value on individuals' responsibility and commitment to one's parents and family [13]. Especially under the cultural context of Confucianism, Chinese adult children have long been shaped by the tenets of filial piety, which emphasize their duties of caring for their parents and fulfilling parents' wishes (even if their wishes are unreasonable at times), and these obligations will be extended as their parents gradually age [14]. Although tourism introduces individuals to "unusual environments" [15], it "often involves connections with, rather than escape from, social relations and the multiple obligations of everyday social life" [16]. Previous studies have indicated that many adult children are unwilling to embark on future trips with their parents because, during such travels, they frequently opt to "constrain themselves and comply with their parents" out of filial piety, sacrificing their own experiences to gratify their parents' desires [17,18]. Consequently, this study aims to investigate how this conflict between the roles of tourists and children influences the perception of family tourism values in the view of adult children. Additionally, it explores how tourist-to-tourist interactions and role conflicts impact adult children's intention to accompany their parents on future trips while such mechanisms remain indistinct. Experiences during these trips may affect the well-being of both the adult children and the parents. In the long term, the trade-off between personal gratification and duty fulfilling can also influence family members' living and development.

Although many researchers have noticed the differences between Eastern tourists and Western tourists [19], research on Chinese family tourism still heavily refers to Western cultural scenarios to understand the psychology and behavior of Chinese family tourists [20,21]. Additionally, most studies in this domain concentrate on families comprising parents and their young children, overlooking the phenomenon of adult children–parents tourism [19,22], which has called for more discussions. Consequently, this study adopts a perspective of tourist-to-tourist interactions and incorporates the theory of role conflict to investigate the following three key questions: (1) Do positive interactions and role conflicts among family members during family tourism influence the formation of family tourism value? (2) How does family tourism value impact adult children's re-travel intention? (3) What specific mechanisms underlie the formation of this intention? This study examines the interplay between positive interaction, role conflict, family tourism values, attitudes, and re-travel intention and explores the potential moderating role of self-efficacy.

At the theoretical level, this study seeks to reveal the process of the formation of "adult children–parents" family tourism value in China, thereby advancing the theoretical understanding of the influence of family tourism value on adult children's re-travel intention. At the practical level, this study can offer enterprises operating family tourism programs valuable insights into the special psychology of adult children who accompany their parents on trips and facilitate a further exploration of the needs of adult children in "adult children–parents" family tourism. Consequently, this might help managers to better identify deficiencies in their tourism products and services, and pinpoint areas for further enhancement and optimization.

## 2. Literature Review

### 2.1. Values of "Adult Children–Parents" Tourism

Previous research has revealed that family tourism yields diverse values depending on the family characteristics. For instance, an investigation into immigrant family tourism has highlighted the significance of preserving the subculture of one's ethnic group as a distinct tourism value esteemed by immigrant families [23]. However, family tourism values are rather different in other relevant research on "children–parent" family tourism. Studies on "young children–parents" family tourism have extensively established the value of such travels in enhancing parental well-being and improving children's academic performance [24,25]. In contrast to family tourism encompassing younger children, family trips involving adolescents with their parents have been found to possess unique value featuring the exploration and formation of the concept of "self", particularly in shaping adolescents' "self-identity" during travel [26]. As children grow up, the evolving awareness of adult children's own norms of responsibility as a "child" propels their cognition of the primary value of traveling with parents to change [27]. Having been relieved from the need to prove their own independence, adult children exhibit a stronger concern for the benefits of these trips to their parents and family. Research on multi-generational family tourism in South Korea indicates that adult children prioritize demonstrating filial piety to their parents during family tourism [8], while other studies, such as the one conducted by Rojas-de-Gracia and Alarcon-Urbistondo on Spanish families, reveal that, differing from children and adolescents [28], adults are less "self-centered" and they tend to focus on the advantages of such trips for their families [29]. Contrary to the previous studies' focus on the "sacrificial role" of adult children in family tourism, there is also research that has discovered that adult children also derive personal happiness from "adult children–parents" family tourism experiences [30].

The disparity in findings necessitates future research to evaluate the value and utility of family tourism from the perspective of adult children so as to clarify their role in "adult children–parents" tourism. Considering the research mentioned above outcomes and the societal context of China's adherence to Confucian culture, this study adopts Wang's classification of the value of adult children accompanying their parents in tourism ("parent-oriented value, family-oriented value, and self-oriented value") [10], categorizing "adult children–parents" family tourism values into three components: the filial piety value of children, the connection value of family, and the personal hedonic value.

### 2.2. Tourist-to-Tourist Interactions and Family Tourism Values

Customers play a key role in the service ecosystem, exerting significant influence on service delivery and consumption through their interactions [31]. The concept of customer-to-customer interaction (CCI) was initially observed by Martin and Pranter, who classified it into two distinct forms: direct interaction, involving interpersonal communication between customers; and indirect interaction, wherein customers are merely part of the scene or service environment [32]. Subsequent research further refined the categorization of customer-to-customer interaction based on various interaction characteristics, such as the subject (stranger interaction and acquaintance interaction) [33], nature (instrumental relationship interaction and emotional relationship interaction), outcome (positive interaction

and negative interaction) [34], and so on. In subsequent research, it has been demonstrated that CCI has an influence on customers' perceived value [35,36], while in tourism research, tourist-to-tourist interactions have also been demonstrated to exert impacts on the formation of tourist experiences, attitudes, and behaviors [37]. As a typical collective consumption scenario [38], tourism activities are considered an ideal environment for tourists to co-create or share service experiences, given their inherent interactive value attributes [39]. Empirical evidence suggests that compared to other leisure activities, family tourism fosters a greater intention to interact between individuals and their family members [22], and people are willing to exert greater efforts to maintain a positive interaction with their family members during family tourism than their daily lives [40]. Differing from tourist-to-tourist interactions among unfamiliar tourists, where much of the existing research focuses [41], tourist-to-tourist interactions among family members are acquaintance interactions, and many of them are direct interactions involving activities such as cooperative preparation [42] and experience sharing [43], which has been proven to yield positive effects in many cases [40,41,44,45]. Therefore, this study proposes the following hypotheses:

**Hypotheses H1a.** *Positive interaction positively impacts filial piety value.*

**Hypotheses H1b.** *Positive interaction positively impacts connection value.*

**Hypotheses H1c.** *Positive interaction positively impacts hedonic value.*

### 2.3. Role Conflicts and Family Tourism Values

Mead adopted the concept of "Role" from drama to explain individual behavior within social contexts [46]. This theoretical framework posits that individuals engage in "Role taking" by considering the expectations and perspectives of others during the process of social interaction, thereby comprehending their social status and subsequently displaying corresponding behavioral patterns. According to Mead's perspective, Linton further elucidates that social roles refer to the expectations, responsibilities, and behaviors assigned to people based upon social status [47]. In contrast to Linton's emphasis on the theory of individuals playing specific roles within a given social structure, Merton argued that individuals take various roles in society, culminating in the formation of a "Role set" [48,49]. Based on this framework, Merton introduced the "Role conflict" theory to elucidate instances where role actors encounter conflicts between the value principles embedded in their respective roles [50].

Role conflict is a theoretical framework that employs theatrical metaphors to expound on individuals' responses to societal and cultural behavioral expectations [51]. Early researchers who utilized the concept of "role" to elucidate social phenomena held different interpretations of this concept, gradually diverging into multiple schools of thought: functionalism, structuralism, symbolic interactionism, organizational perspectives, and cognitive role theory [52]. Previous research in tourism has predominantly focused on role conflicts that arise within occupational contexts, such as those experienced by employees in travel agencies [53–57], hotels [58,59], and tour guides [60,61]. Consequently, many of these studies adopted the framework of organizational perspectives to elucidate role conflicts within tourism activities and lacked exploration regarding potential role conflicts among tourists themselves.

Previous research has posited that family tourists may encounter an "implicit" role conflict, predominantly stemming from self-alienation induced by self-sacrifice. Though this type of role conflict may not be as overt as explicit role conflicts, it often exerts a greater impact [40]. In "adult children–parents" family tourism, it is adult children who often assume the role of self-sacrifice. When adult children travel with their parents, their role as "children" has already been pre-established long before their travel. Consequently, this leads to profound influence from societal norms and filial piety expectations during the trip. Furthermore, the inevitable interactions between parents and children within the

shared tourist environment continually shape the tourist roles assumed by both parties [62]. From the view of adult children, when the expectations of the tourist role, involving leisure and enjoyment, conflict with the expectations of the child role, requiring filial piety and obedience (e.g., differences of opinion, behavioral conflicts, etc.) [63], the internal role conflict within the adult children may rise, which causes a negative family tourism consequence [64]. Therefore, this study proposes the following hypotheses:

**Hypotheses H2a.** *Role conflict negatively impacts filial piety value.*

**Hypotheses H2b.** *Role conflict negatively impacts connection value.*

**Hypotheses H2c.** *Role conflict negatively impacts hedonic value.*

*2.4. Moderating Role of Self-Efficacy on the Formulation of Family Tourism Values*

Self-efficacy, a concept rooted in social cognitive theory, refers to individuals' beliefs in their capacity to perform specific behaviors or attain particular achievements [65]. Research in human–robot interaction indicates that cases where tourists exhibit a higher level of self-efficacy, which signifies greater confidence in their own abilities, will consequently lead to better interactive experiences [66]. Within the context of interpersonal interactions, self-efficacy specifically denotes individuals' judgment and confidence in their abilities to engage with others [67], encompassing behaviors such as expressing confidence, offering and accepting assistance, and resolving interpersonal conflicts [68]. Previous research has corroborated a moderating effect of self-efficacy on the formation of experiential value, that is, customers with higher self-efficacy can better understand their role, are more inclined to enhance their capacity for value co-creation, and believe in their ability to surmount obstacles and overcome difficulties [69]. Therefore, this study proposes the following hypotheses:

**Hypotheses H3a.** *Self-efficacy moderates the impact of positive interaction on filial piety value.*

**Hypotheses H3b.** *Self-efficacy moderates the impact of positive interaction on connection value.*

**Hypotheses H3c.** *Self-efficacy moderates the impact of positive interaction on hedonic value.*

**Hypotheses H3d.** *Self-efficacy moderates the impact of role conflict on filial piety value.*

**Hypotheses H3e.** *Self-efficacy moderates the impact of role conflict on connection value.*

**Hypotheses H3f.** *Self-efficacy moderates the impact of role conflict on hedonic value.*

*2.5. Family Tourism Values, Attitude, and Re-Travel Intention*

In the early stages of research within the field of social psychology, scholars across various research directions have employed diverse models to elucidate the mechanisms behind individuals' decision-making behaviors. Examples of such models include the theory of reasoned action (TRA), the theory of planned behavior (TPB), and the cognition–attitude–behavior (CAB) model. These investigations have substantiated that "attitude" is a belief formed by individuals through the evaluation of objects/behaviors [70], which in turn exerts an influence on their behavioral mechanisms. Building upon this foundation, Feather's research revealed intricate interrelationships among individuals' values, attitudes, and behaviors [71]. Subsequently, Homer and Kahle devised the value–attitude–behavior (VAB) model to explain social behaviors [72], establishing that an individual's value judgments can influence their behavioral intentions through attitudes [73]. Presently, the VAB model finds widespread application in explicating consumer behaviors within various fields including tourism industry [74–76]. In the VAB model, value is an abstract

and most profound cognition [77,78] that refers to a fundamental standard utilized by customers when making a purchasing decision [79]. In the context of this study, the concept corresponding to values pertains to family tourism values, encompassing filial piety value, family connection value, and hedonic value. Attitude is ubiquitous in research on individuals' behaviors [80]. In this study, attitude denotes the adult children's perspectives and evaluations of participating in travel activities with their parents, while behavior is represented by adult children's re-travel intention.

Therefore, this study proposes the following hypotheses:

**Hypotheses H4a.** *Filial piety value positively impacts attitude.*

**Hypotheses H4b.** *Connection value positively impacts attitude.*

**Hypotheses H4c.** *Hedonic value positively impacts attitude.*

**Hypotheses H5.** *Attitude positively impacts adult children's re-travel intention.*

Based on the above literature review and study hypothesis, the conceptual framework of this research has been deduced, as shown in Table 1 and Figure 1.

**Table 1.** Summary of hypotheses.

| Hypotheses | Description |
|---|---|
| H1a–H1c | To examine whether positive interaction impacts family tourism values and to ascertain the magnitude of these impacts. |
| H2a–H2c | To examine whether role conflict impacts family tourism values and to ascertain the magnitude of these impacts. |
| H3a–H3c | To examine whether self-efficacy moderates the impact of positive interaction on family tourism values. |
| H3d–H3f | To examine whether self-efficacy moderates the impact of role conflict on family tourism values. |
| H4a–H4c | To examine whether family tourism values impact attitude and to ascertain the magnitude of these impacts. |
| H5 | To examine whether attitude impacts adult children's re-travel intention. |

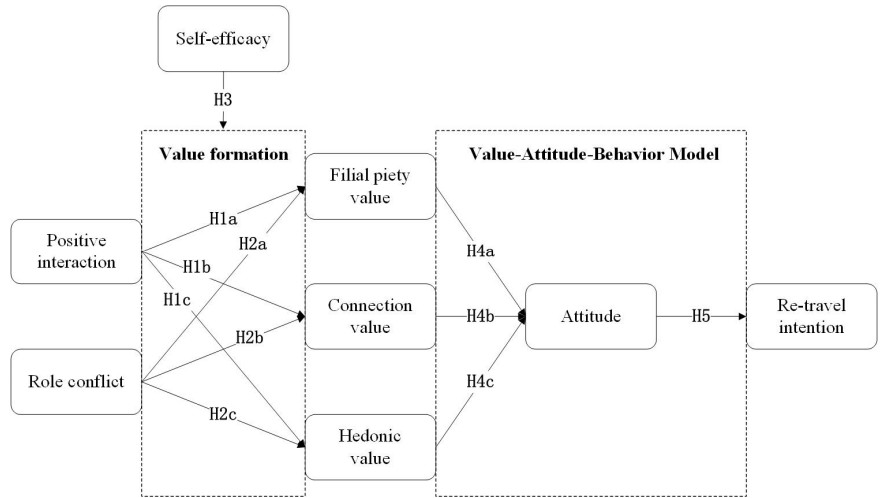

**Figure 1.** Conceptual framework.

## 3. Methods

### 3.1. Survey Instrument

The measurement of this study includes eight dimensions: positive interaction, role conflict, self-efficacy, filial piety value, family connection value, hedonic value, attitude,

and re-travel intention, and all measurement scales are 5-point Likert scales, where 1 means strongly disagree/always, and 5 means strongly agree/never. The measurement scales of positive interaction, attitude, re-travel intention, hedonic value, and family connection value were from multiple researchers' works [81–84]. Items of role conflict and self-efficacy were adapted from the measurement scales of Rizzo et al. and Chen et al. [85,86]. Measurement scale of filial piety value is designed according to the research conclusions of Fu, Li, and Yan [87]. The demographic characteristics section of the questionnaire was designed based on existing research on family tourism, including 6 items: gender, age, education level, occupation, income, marriage. A total of 92 samples were collected in a pilot study prior to the formal survey. The results showed that all Cronbach's alpha coefficients for eight variables were greater than 0.80, and exploratory factorial analysis showed a KMO = 0.865 and a significant Bartlett's sphericity test, suggesting that the questionnaire had good reliability and validity.

### 3.2. Data Collection

This study aimed to survey adult children in the Jiangsu, Zhejiang and Shanghai regions, from 1 May 2023 to 14 May 2023. A professional offline survey company, Wenjuan.com, was used to physically invite people to scan the QR code linked to questionnaires and fill out the questionnaires. Meanwhile, in order to mitigate the common method bias that might stem from a singular data source, as in previous research [88,89], an offline survey was also employed by using professional online survey company Credamo. Though research using Credamo to collect data has been published in many top tourism journals [90], considering that online survey companies often use respondent pools to collect data, the data they provide are considered more susceptible to representational biases compared to data collected by offline survey. Therefore, the proportion of online research data in the analysis of this study is controlled within a relatively narrow range (24.03%), primarily employed for testing data consistency.

To ensure the validity of the sample, the survey took three measures during the formal survey: (1) declaring that all data in this survey would only be used for this academic research, ensuring the objectivity and neutrality of the answers; (2) setting questionnaire screening questions and controlling IP sources to ensure that questionnaire respondents were preset survey subjects; (3) excluding questionnaires with response times less than 210 s. Also, this study contains diverse data collection methods, so duplicate responses from the same participants may exist. In previous research, some studies employed a methodology that combines demographic characteristic with other characteristics of individuals, such as postal codes/email addresses, to discern whether there are the same participants [91,92]. Additionally, studies that have employed online questionnaires (participants can be invited both online or physically) will discern that there are the same participants based on their IP addresses [93,94]. Given the conformity of the methodological approach, this study combines demographic characteristic with IP address to discern whether there are the same participants. As a consequence, a stricter data-screening procedure was performed to ascertain if repeated questionnaires demonstrating both identical response locations (cities determined based on IP address) and demographic characteristics existed, and no instances of such were found. Moreover, given that the research encompassed 26 cities with a total population of 2.27 billion, the likelihood of encountering the same participants can be considered negligible. Although this is a study on family tourism, it differs from typical family tourism research due to its focus on adult children's perspectives. All participants in the survey are adults aged over 18 and have accompanied their parents on trips before, so it does not involve special groups such as the elderly individuals or young children. Questionnaire respondents were fully informed about the information they would provide and its purpose. Therefore, this study does not encounter serious research ethics problems. After completing the questionnaire, online respondents would receive USD 0.29, while offline respondents would receive USD 1.01.

A total of 648 questionnaires were collected in the formal survey, and 82 questionnaires with consistent answers, missed answers, and invalid IP addresses were excluded. Therefore, 566 valid questionnaires were selected, with an effective recovery rate of 87.35%.

### 3.3. Data Analysis

This research employed multiple data analytical methodologies, including exploratory factor analysis (EFA), confirmatory factor analysis (CFA), structural equation modeling (SEM), and various statistical software tools, encompassing SPSS 20.0 and AMOS 24.0, were utilized in the analytical procedure. Specifically, a systematic analytical approach was adopted, commencing with the examination of potential common method bias within the dataset. This involved an initial application of CFA to assess the validity of a theoretically derived measurement model, followed by an evaluation of hypothesized relationships upon achieving a construct model deemed satisfactory. Moreover, the study empirically examined the presence of mediating effects, moderating effects, and moderated mediating effects within the conceptual model through the utilization of the bootstrapping method.

## 4. Results

### 4.1. Overview of the Sample Population

The sample structure is as follows (Table 2). In terms of gender, 244 are males, accounting for 43.11%, and 322 are females, accounting for 56.89%. When it comes to other key items, such as age, monthly income, and education level, all measurement variables basically follow a normal distribution. Thus, the distribution of the sample is considered adequate and demonstrates good representativeness.

**Table 2.** Socio-demographic characteristics of the main survey respondents ($N = 566$).

| | Variable | Frequency | (%) |
|---|---|---|---|
| Gender | Male | 244 | 43.11 |
| | Female | 322 | 56.89 |
| Marital Status | Single | 73 | 12.90 |
| | Unmarried but have a girlfriend/boyfriend | 85 | 15.02 |
| | Married | 406 | 71.73 |
| | Divorced | 2 | 0.35 |
| Education | High school/special school/technical school | 67 | 11.84 |
| | Two-year college/Four-year university | 434 | 76.68 |
| | Master's degree or above | 65 | 11.48 |
| Monthly Income (USD) | <723.5 | 69 | 12.19 |
| | 723.5–1447 | 233 | 41.17 |
| | 1447–2894 | 212 | 37.46 |
| | >2895 | 52 | 9.19 |
| Occupation | Enterprise management personnel | 107 | 18.90 |
| | Enterprise employees and self-employed individuals | 310 | 54.77 |
| | Freelancer | 43 | 7.60 |
| | Public officials | 49 | 8.66 |
| | Retired | 4 | 0.71 |
| | Student | 51 | 9.01 |
| | Other | 2 | 0.35 |
| Age | 18–24 | 83 | 14.66 |
| | 25–34 | 281 | 49.65 |
| | 35–44 | 156 | 27.56 |
| | ≥45 | 46 | 8.13 |

### 4.2. Common Method Variance Test

Harman's one-factor test was used to investigate the existence of potential common method bias in the data, followed by an unrotated exploratory factor analysis. The findings revealed that eight factors with eigenvalues greater than 1 were identified, and the maximum factor variance explained was 30.973%, lower than the 50% threshold. Therefore,

it may be concluded that the data fulfil the requirements for the assessment of common method bias. Furthermore, Table 3 displays the fit indices for the single-factor confirmatory factor analysis. All of the model fit indices indicate a poor fit, verifying the lack of a common single factor in the research data. As a result, both methods mentioned above have verified the absence of any significant common method bias in this study.

**Table 3.** Evaluation result of the fitting effect of single-factor confirmatory factor analysis.

| The Fit Degree of Integral Model | Fit Index | Numerical Value | Fit Criterion | Literature Source |
|---|---|---|---|---|
| Absolute fit index | $\chi^2/\mathrm{df}$ | 9.654 | 1~3 | Carmines and Meiver [95] |
| | RMR | 0.106 | ≤0.05 | Byrne [96] |
| | RMSEA | 0.124 | ≤0.1 | Hoyle and Panter [97] |
| | AGFI | 0.548 | ≥0.8 | Sharma [98] |
| | GFI | 0.607 | ≥0.8 | Robert [99] |
| Relative fit index | NFI | 0.533 | ≥0.9 | |
| | IFI | 0.561 | ≥0.9 | Bentler and Bonett [100] |
| | TLI | 0.526 | ≥0.9 | Hu and Bentler [101] |
| | CFI | 0.559 | ≥0.9 | |

*4.3. Validity and Reliability Test for Measures*

This study utilized an exploratory factor analysis as a methodological strategy for examining the fundamental dimensional framework of the complete item set. The analytical results reveal a Kaiser–Meyer–Olkin (KMO) value of 0.920, surpassing the recommended threshold of 0.7, and a Bartlett's sphericity test *p*-value of 0.000, which is below the conventional significance level of 0.001 [102].

Collectively, these results affirm the suitability of the data for factor analysis. A total of 30 items were retained, resulting in 8 factors emerging that collectively explained 66.31% of the cumulative variance contribution rate, surpassing the threshold of 60% [103], and no items were excluded. Subsequent examination of Table 4 reveals salient factor loadings, Cronbach's alpha (α) values, average variance extracted (AVE) scores, and construct reliability (CR) values.

**Table 4.** Results of exploratory factor analysis.

| Constructs and Items | Factor Loading |
|---|---|
| *Positive Interaction (Cronbach's α = 0.748, CR = 0.75, AVE = 0.501)* | |
| Communicate interesting things about the journey with parents | 0.762 |
| Help and care for each other with parents during the trip | 0.664 |
| Have shared memories with parents (such as taking photos and taking a group photo) | 0.693 |
| *Role conflict (Cronbach's α = 0.905, CR = 0.905, AVE = 0.656)* | |
| Some of my parents' behavioral habits conflict with me (such as frugality, paternalism, uncivilized behavior) | 0.838 |
| Some of my parents' requirements are inconsistent with my wishes (such as choosing tourist attractions and making activity decisions) | 0.780 |
| Some of my parents' demands make me feel unhappy | 0.808 |
| Some of my parents' demands make me feel unnecessary | 0.806 |
| Some of my parents' demands made me feel at a loss | 0.816 |
| *Self-efficacy (Cronbach's α = 0.775, CR = 0.774, AVE = 0.534)* | |
| I have the ability to positively interact with my parents | 0.732 |
| I can overcome the difficulty of interacting with my parents | 0.785 |
| Overall, I believe that my interaction with parents can achieve my anticipation | 0.753 |
| *Filial piety Value (Cronbach's α = 0.81, CR = 0.81, AVE = 0.515)* | |
| Through family tourism, I accompany and take care of my parents | 0.725 |
| Through family tourism, I repay my parents and show them filial piety | 0.69 |
| Through family tourism, I satisfy my parents | 0.724 |
| Through family tourism, I fulfill my parents' wishes | 0.732 |

**Table 4.** *Cont.*

| Constructs and Items | Factor Loading |
|---|---|
| *Connection Value (Cronbach's α = 0.776, CR = 0.777, AVE = 0.537)* | |
| Spent a wonderful time together | 0.758 |
| Obtained a shared experience | 0.720 |
| Formed shared memories | 0.720 |
| *Hedonic Value (Cronbach's α = 0.762, CR = 0.763, AVE = 0.518)* | |
| Enables me to escape from the mundane affairs of life temporarily | 0.737 |
| Enables me to relieve myself from the pressure of daily life temporarily | 0.726 |
| Enables me to put aside my daily troubles temporarily | 0.737 |
| *Attitude (Cronbach's α = 0.858, CR = 0.859, AVE = 0.549)* | |
| I think traveling with my parents is worthwhile | 0.759 |
| I think traveling with my parents is meaningful | 0.743 |
| I think traveling with my parents is a good choice | 0.715 |
| I think it's a good idea to travel with my parents | 0.759 |
| I am in favor of traveling with my parents | 0.729 |
| *Re-travel Intention (Cronbach's α = 0.888, CR = 0.888, AVE = 0.664)* | |
| I am willing to travel with my parents again | 0.840 |
| I am very likely to travel with my parents again | 0.815 |
| I will continue to travel with my parents in the future | 0.828 |
| Traveling with parents again will be worthwhile | 0.775 |

Note: CFA model fits: $\chi^2/df$ = 1.510, RMR = 0.027, RMSEA = 0.03, NFI = 0.932, IFI = 0.976, TLI = 0.972, CFI = 0.976, GFI = 0.937, AGFI = 0.923. The italics in the table are the reliability and validity measurement indicators of each variable.

As is shown in Table 4, both Cronbach's alpha (α) and CR values associated with the observed variables exceeded the threshold of 0.7 [103] and 0.6 [104], underscoring the high reliability of the measurement scale. Furthermore, the AVE scores for the observed variables exceeded 0.5 [103], while the standardized factor loadings for each item within the scale exhibited a range of 0.664 to 0.840, all surpassing the 0.5 threshold [105] and achieving statistical significance at a specified level of significance (*p*-value), indicating that the measurement model has good aggregation validity.

Additionally, the outcomes of the differential validity assessment, as presented in Table 5, manifest that the square root of the AVE for each latent variable surpassed its correlation coefficient with other latent variables, indicating that the measurement scale has good discriminative validity [106].

**Table 5.** Square root of AVE and correlation coefficient.

| Constructs | 1 | 2 | 3 | 4 | 5 | 6 | 7 | 8 |
|---|---|---|---|---|---|---|---|---|
| 1 | **0.708** | | | | | | | |
| 2 | −0.284 | **0.810** | | | | | | |
| 3 | 0.32 | −0.298 | **0.731** | | | | | |
| 4 | 0.482 | −0.375 | 0.551 | **0.718** | | | | |
| 5 | 0.514 | −0.409 | 0.522 | 0.656 | **0.733** | | | |
| 6 | 0.486 | −0.238 | 0.362 | 0.329 | 0.388 | **0.720** | | |
| 7 | 0.374 | −0.382 | 0.387 | 0.718 | 0.69 | 0.323 | **0.741** | |
| 8 | 0.315 | −0.262 | 0.342 | 0.6 | 0.48 | 0.223 | 0.706 | **0.815** |

Note: 1 = positive interaction; 2 = role conflict; 3 = self-efficacy; 4 = filial piety value; 5 = connection value; 6 = hedonic value; 7 = attitude; 8 = re-travel intention; diagonal values (bold) are AVE values. Off-diagonal values (plain) were squared inter-construct correlations of the constructs.

### 4.4. Structure Model and Hypothesis Testing

In this study, the fit indices ($\chi^2/df$ = 1.774, RMR = 0.036, RMSEA = 0.037, NFI = 0.927, IFI = 0.967, TLI = 0.963, CFI = 0.967, GFI = 0.933, AGFI = 0.919) indicate that the structural model fit well to the data (Table 6). Regarding the relationship between the variables in the model, as shown in Table 6, all hypotheses are supported except for H4c.

**Table 6.** Evaluation result of the fitting effect of the confirmatory factor analysis.

| The Fit Degree of Integral Model | Fit Index | Numerical Value | Fit Criterion | Literature Source |
|---|---|---|---|---|
| Absolute fit index | $\chi^2/\mathrm{df}$ | 1.774 | 1~3 | Carmines and Meiver [95] |
| | RMR | 0.036 | $\leq 0.05$ | Byrne [96] |
| | RMSEA | 0.037 | $\leq 0.1$ | Hoyle and Panter [97] |
| | AGFI | 0.919 | $\geq 0.8$ | Sharma [98] |
| | GFI | 0.933 | $\geq 0.8$ | Robert [99] |
| Relative fit index | NFI | 0.927 | $\geq 0.9$ | |
| | IFI | 0.967 | $\geq 0.9$ | Bentler and Bonett [100]; |
| | TLI | 0.963 | $\geq 0.9$ | Hu and Bentler [101] |
| | CFI | 0.967 | $\geq 0.9$ | |

Specifically, positive interaction has a significant positive effect on filial piety value ($\beta = 0.465$, $p < 0.001$), connection value ($\beta = 0.495$, $p < 0.001$), and hedonic value ($\beta = 0.485$, $p < 0.001$), while role conflict has a significant negative effect on filial piety value ($\beta = -0.257$, $p < 0.001$), connection value ($\beta = -0.282$, $p < 0.001$) and hedonic value ($\beta = -0.101$, $p < 0.05$). However, only filial piety value ($\beta = 0.526$, $p < 0.001$) and connection value ($\beta = 0.391$, $p < 0.001$) have a significant positive effect on attitude, while hedonic value does not exert a significant effect on attitude ($\beta = 0.010$, $p = 0.810$). Finally, attitude impacts adult children's re-travel intention significantly ($\beta = 0.708$, $p < 0.001$). The concrete results are shown in Table 7.

**Table 7.** Standardization path coefficient and hypothesis testing results.

| Hypothesis | $\beta$ | SE | *t*-Value | *p*-Value | Result |
|---|---|---|---|---|---|
| H1a: PI → FV | 0.465 | 0.075 | 7.741 | *** | Supported |
| H1b: PI → CV | 0.495 | 0.075 | 8.213 | *** | Supported |
| H1c: PI → HV | 0.485 | 0.073 | 7.864 | *** | Supported |
| H2a: RC → FV | −0.257 | 0.031 | −5.369 | *** | Supported |
| H2b: RC → CV | −0.282 | 0.031 | −5.808 | *** | Supported |
| H2c: RC → HV | −0.101 | 0.03 | −2.008 | * | Supported |
| H4a: FV → AT | 0.526 | 0.051 | 9.354 | *** | Supported |
| H4b: CV → AT | 0.391 | 0.05 | 7.028 | *** | Supported |
| H4c: HV → AT | 0.01 | 0.041 | 0.24 | 0.810 | Unsupported |
| H5: AT → RI | 0.708 | 0.067 | 14.539 | *** | Supported |

Note: PI = positive interaction; RC = role conflict; FV = filial piety value; CV = connection value; HV = hedonic value; AT = attitude; RI = re-travel intention; *** $p < 0.001$; * $p < 0.05$.

The hypothesis H4c within the conceptual framework is not supported. One plausible explanation for this is that hedonic value does not rank as paramount for adult children within the context of "adult children–parents" family tourism, different from the situation when adult children are independent travelers.

In summary, the hypothesis testing results of this study's theoretic framework are as follows (Figure 2).

### 4.5. Test of Mediating Effect

Hayes (2013) highlighted that the Bootstrap mediation effect test method exhibits robust data applicability and statistical performance when juxtaposed with conventional approaches such as the Sobel test [107]. Consequently, this investigation employed the Bootstrap mediation effect test within AMOS to conduct 5000 repeated sampling tests to examine the mediating role of family tourism values and attitudes between positive interactions, role conflicts, and re-travel intention. The test results show that the coefficients of the four paths do not include zero in the 95% confidence intervals of bias-corrected and percentile, which means the mediating effect is significant (Table 8).

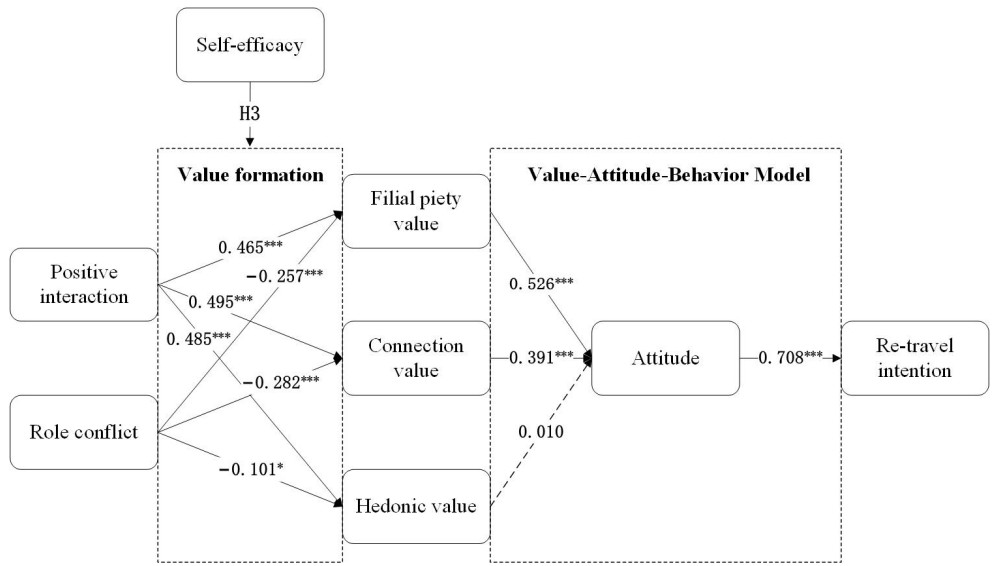

**Figure 2.** Outcomes of hypotheses testing. *** *p* < 0.001; * *p* < 0.05.

**Table 8.** Bootstrap test result of mediating effect (standardized coefficient).

| Path | β | SE | Bias-Corrected 95%CI | | | Percentile 95%CI | | |
|---|---|---|---|---|---|---|---|---|
| | | | Lower | Upper | *p* | Lower | Upper | *p* |
| PI → FV → AT → RI | 0.173 | 0.038 | 0.112 | 0.265 | *** | 0.107 | 0.255 | *** |
| PI → CV → AT → RI | 0.137 | 0.036 | 0.079 | 0.222 | *** | 0.076 | 0.217 | *** |
| RC → FV → AT → RI | −0.096 | 0.027 | −0.156 | −0.049 | *** | −0.153 | −0.047 | *** |
| RC → CV → AT → RI | −0.078 | 0.024 | −0.133 | −0.039 | *** | −0.129 | −0.036 | *** |

Note: PI = positive interaction; RC = role conflict; FV = filial piety value; CV = connection value; HV = hedonic value; AT = attitude; RI = re-travel intention; *** *p* < 0.001.

### 4.6. Test of Moderating Effect

Ping introduced the utilization of a multi-indicator approach with interactions to examine the moderating effects of latent variables, which has gained widespread acknowledgment. Thus, this study adopts Ping's approach to conduct an in-depth analysis of the moderating role of self-efficacy. Interactions for self-efficacy and positive interaction, as well as role conflict, are constructed, and both of these interactions are incorporated into the model. The fit indices for the new model are as follows: $\chi^2/\mathrm{df}$ = 1.556, RMR = 0.028, RMSEA = 0.031, NFI = 0.907, IFI = 0.964, TLI = 0.96, CFI = 0.964, GFI = 0.921, AGFI = 0.906, indicating that the structural model fits well to the data. Also, the two interactions exert significant effects on the three influence paths between positive interaction, role conflict and filial piety value, family connection value, and hedonic value (Table 9).

**Table 9.** Results of the moderating test (standardized coefficient).

| Hypothesis | β | SE | *t*-Value | *p*-Value | Result |
|---|---|---|---|---|---|
| H3a: SE*PI → FV | 0.208 | 0.11 | 2.436 | * | Supported |
| H3b: SE*PI → CV | 0.295 | 0.121 | 3.133 | ** | Supported |
| H3c: SE*PI → HV | 0.195 | 0.103 | 2.29 | * | Supported |
| H3d: SE*RC → FV | 0.247 | 0.111 | 2.828 | ** | Supported |
| H3e: SE*RC → CV | 0.353 | 0.118 | 3.747 | *** | Supported |
| H3f: SE*RC → HV | 0.198 | 0.103 | 2.291 | * | Supported |

Note: PI = positive interaction; RC = role conflict; FV = filial piety value; CV = connection value; HV = hedonic value; SE = self-efficacy; SE*PI represents the interaction of self-efficacy and positive interaction; SE*RC represents the interaction of self-efficacy and role conflict; *** *p* < 0.001; ** *p* < 0.01; * *p* < 0.05.

Also, to more intuitively demonstrate the regulatory effects of self-efficacy, this study applies the recommendations of Aiken and West's advice [108], drawing a simple slope test for the analysis (Figure 3). As shown in Figure 3, a lower self-efficacy caused a reduced family tourism value and exacerbates the adverse impact of role conflict on the family tourism. Conversely, a higher self-efficacy amplifies the family tourism value and mitigates the negative effect of role conflict. When the self-efficacy of adult children is high enough, the influence of role conflict on personal hedonic values will transform from negative to positive.

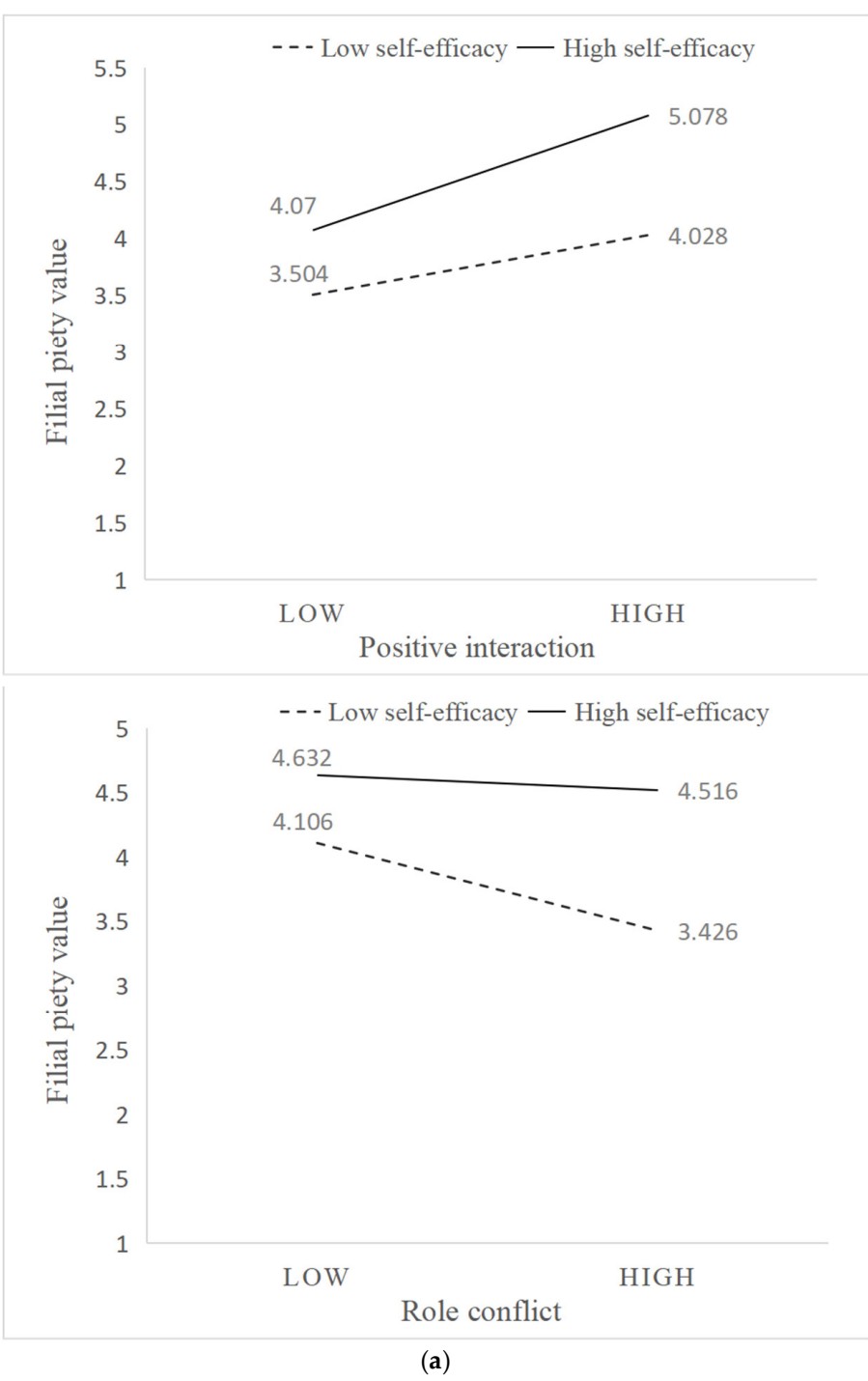

(a)

**Figure 3.** *Cont.*

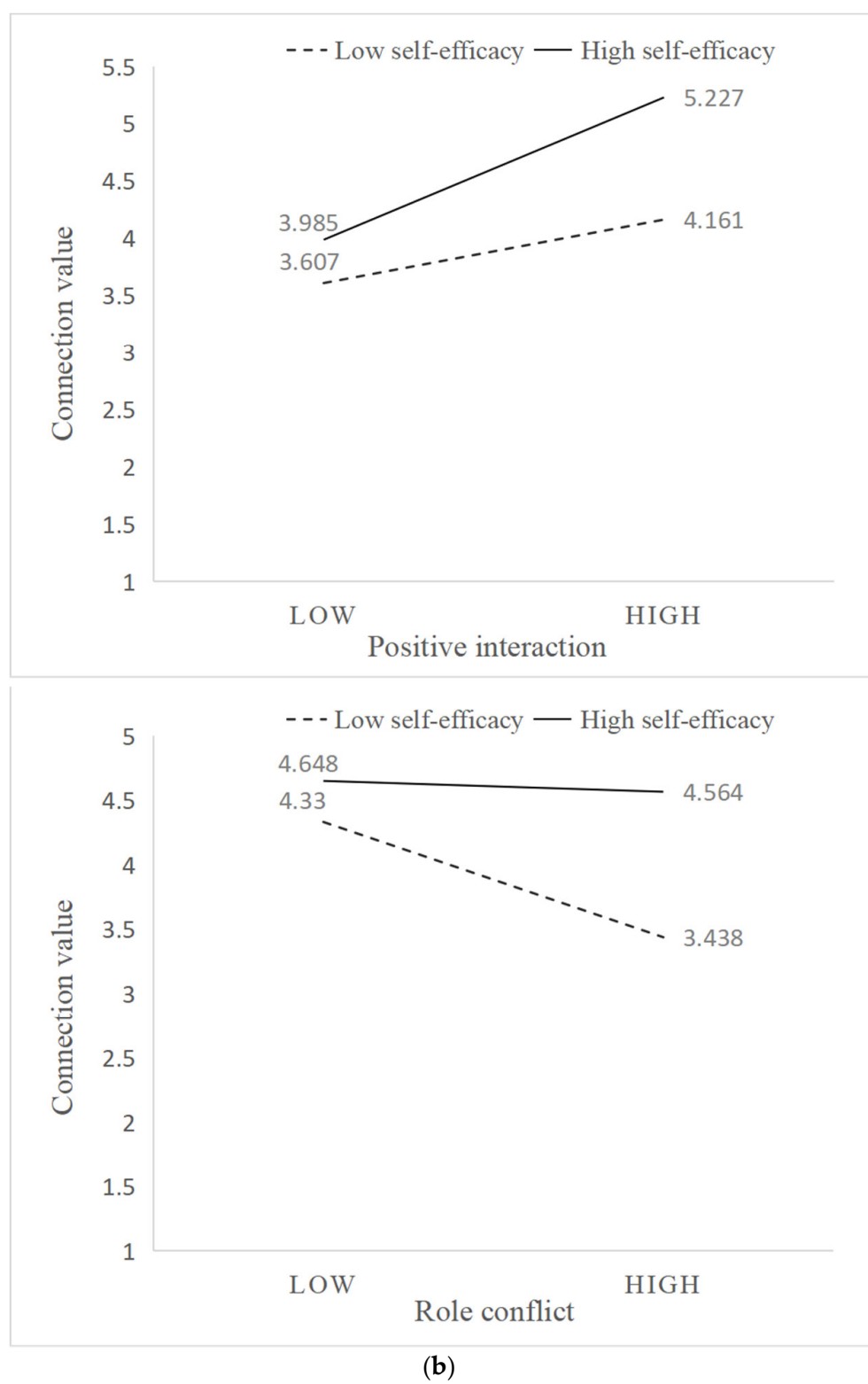

**Figure 3.** *Cont.*

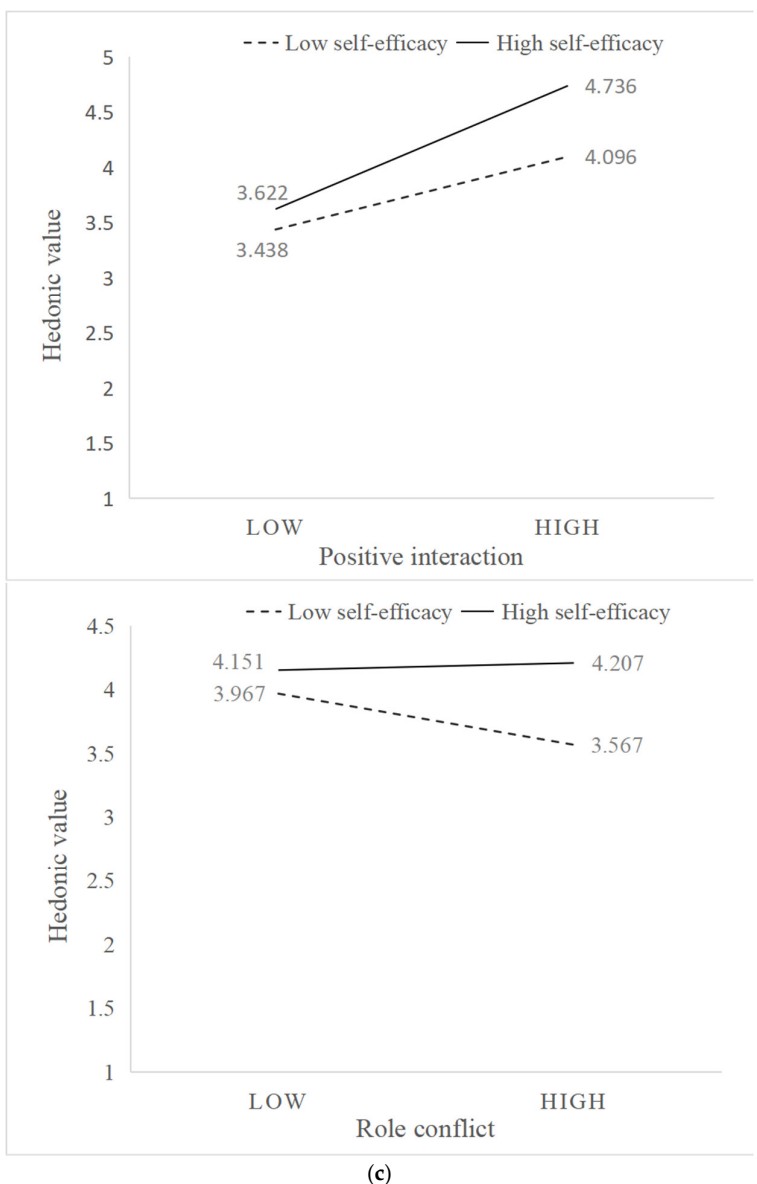

**Figure 3.** The moderating effect of self-efficacy. (**a**) The moderating effect of self-efficacy on the formation of filial piety value. (**b**) The moderating effect of self-efficacy on the formation of connection value. (**c**) The moderating effect of self-efficacy on the formation of hedonic value.

This phenomenon may stem from the capacity of adult children who have higher self-efficacy to adopt a more flexible posture to handle the conflicts with their parents, employing tactics such as "persuasion coupled with persistence" and "benevolent falsehoods", thus successfully realizing their goals and improving their hedonic value. This observation corroborates previous research conclusions [45], affirming that when instances of value co-destruction arise within the context of family tourism, adult children can adopt appropriate coping strategies to guide the transformation from value co-destruction to value co-creation, ultimately resulting in value restoration.

### 4.7. Test of Moderated Mediation Effect

This study has examined the existence of chain mediation and moderating effects in the model. Consequently, a moderated chain mediation effect needs to be tested. The test results in Figure 3 and Table 10 show that although the self-efficacy of adult children plays a moderating role in shaping the three family tourism values, it is only the influence the high self-efficacy exerts on the formation of family connection value that impacts the

adult children's re-travel intention. This phenomenon can be attributed to the fact that the elevation of filial piety value brought by adult children's high self-efficacy often represents a psychological readiness for concessions and compromises when conflicts arise between the children and parents during the family tourism. This psychological activity will contribute to the formation of filial piety value but will not further help improve adult children's re-travel intention. When it comes to hedonic value, as mentioned above, it is not the foremost concern of adult children within the context of "adult children–parents" family tourism, and thus, the augmentation of hedonic value brought by adult children's high self-efficacy does not naturally bear an effect on their re-travel intention.

**Table 10.** Bootstrap test result of moderated chain mediation effect (standardized coefficient).

| Path | β | SE | Bias-Corrected 95%CI | | | Percentile 95%CI | | |
|---|---|---|---|---|---|---|---|---|
| | | | Lower | Upper | *p* | Lower | Upper | *p* |
| SE*PI → FV → AT → IT | 0.044 | 0.221 | −0.006 | 0.691 | 0.099 | −0.005 | 0.691 | 0.099 |
| SE*PI → CV → AT → IT | 0.047 | 0.196 | 0.004 | 0.579 | * | 0.005 | 0.585 | * |
| SE*RC → FV → AT → IT | 0.098 | 0.301 | −0.025 | 0.712 | 0.144 | −0.01 | 0.949 | 0.08 |
| SE*RC → CV → AT → IT | 0.105 | 0.269 | 0.017 | 0.63 | * | 0.026 | 0.805 | ** |

Note: PI = positive interaction; RC = role conflict; FV = filial piety value; CV = connection value; HV = hedonic value; AT = attitude; RI = re-travel intention; SE = high self-efficacy; ** $p < 0.01$; * $p < 0.05$.

## 5. Discussion

### 5.1. Theoretical Implications

Drawing upon the theoretical frameworks of role conflict theory, this study aims to explore the influence of tourist-to-tourist interactions by adopting the "value–attitude–behavior" model to reveal the mechanism of family tourism values on adult children's re-travel intention. An empirical investigation was conducted with 566 adult children from Jiangsu, Zhejiang, and Shanghai in this study. Through hypothesis testing of the conceptual model, the study yields the following four key conclusions.

Firstly, positive interaction between adult children and their parents during family tourism has a positive impact on forming family tourism values. This conclusion is consistent with several relevant research. Specifically, it establishes that a higher frequency of positive interaction contributes to a heightened perception of family tourism value among adult children. This conclusion aligns with some previous findings, emphasizing the influence of tourist-to-tourist interactions on tourism values [81]. Also, it verified some studies' conclusions that recognized family travel experiences' function of creating multifaceted values [109]. Through quantitative analysis, this study further discerns three distinct components constituting the value of "adult children–parents" family tourism: the "filial piety value of children, connection value of family, and personal hedonic value", thus corroborating the research findings of Wang et al. [8].

Secondly, role conflict has a negative influence on the formation of family tourism values. The adverse ramifications of tourists' conflicts on the tourist experience have been extensively substantiated [110], and similar cases are also observed in family tourism [39]. However, previous studies concerning family tourism conflicts predominantly concentrate on overt behavioral conflicts, exemplified by disputes and verbal altercations. Evidently, although transpiring within the context of family tourism, such conflicts remain essentially inter-tourist conflicts akin to disputes, selfish conduct, and etiquette lapses among unfamiliar tourists during package tours [111]. Consequently, this explanation is not able to elucidate the intrinsic attributes and disparities unique to "adult children–parents" family tourism. Also, several studies have endeavored to explain these matters through the lens of "filial piety" [17], and indeed, they have offered some insightful explanations for adult children's motivations to travel with their parents, as well as direct causes of the conflicts between adult children and their parents, such as "divergent preferences, cognitive disparities, and conceptual distinctions". Yet, they are still not able to reveal the root causes

of these intergenerational conflicts and differentiating factors between such conflicts and those arising among unfamiliar tourists. The conclusion that role conflict has a negative impact on family tourism values indicates that while traveling with their parents, adult children have to concurrently take the roles of both a "child" and a "tourist", inevitably precipitating conflicts between these dual roles. Although there are certain commonalities in behavioral manifestations between conflicts among unfamiliar tourists and "child–tourist" role conflict experienced by adult children, profound disparities can be found in the formation mechanisms of these two categories of conflicts. The former pertains to conflicts in interactive behavior within the shared tourist space, while the latter emanates from conflicts of personal volition resulting from the superposition of the "family context" and the "tourism context". This result unveils that family tourism is fundamentally disparate from other genres of tourism, amalgamating the dual attributes of the "usual environment" of the "family context" and the "unusual environment" of the "tourism context".

Thirdly, positive interaction and role conflict initially impact the filial piety value, connection value, and hedonic value, subsequently shaping their attitudes and ultimately influencing adult children's re-travel intention. Existing research views the relationship between family tourism interactions and re-travel intention differently. For instance, some research posits that re-travel intention is a direct consequence of family tourism interactions, while other research suggests that re-travel intention is a sequential development result of various tourist-to-tourist interactions with causal relationships, denoted as a chain process of "tourism preparation → experiential process → evaluative appraisal → re-travel intention" [18]. A theoretical contribution of this investigation lies in its elucidation of the intricate mechanism by which family tourism interaction reverberates onto adult children's re-travel intention, thereby confirming that the mechanism is a chain-acting process. This analysis provides empirical substantiation to the intricate trajectory of "family tourism interaction → family tourism values → attitude → re-travel intention". Furthermore, this study refines the family tourism values into three components: "filial piety value of children, connection value of family, and personal hedonic value", thus conducting a deeper exploration into the distinct avenues by which affirmative interaction and role conflicts impinge upon adult children's re-travel intention.

Additionally, another significant finding of this study is the varying impact of the size of the influence effects on family tourism values and their influence on re-travel intention. An intriguing finding emerges from Table 7, where it can be discerned that, despite positive interaction having nearly equivalent effects on filial piety value ($\beta = 0.465$, $p < 0.001$), family connection value ($\beta = 0.495$, $p < 0.001$), and hedonic value ($\beta = 0.485$, $p < 0.001$), role conflict exhibits a notable disparity in its negative impact across these values. To be more specific, the influence of role conflict on hedonic value ($\beta = -0.101$, $p < 0.05$) is considerably less pronounced than that on filial piety value ($\beta = -0.257$, $p < 0.001$) and family connection value ($\beta = -0.282$, $p < 0.001$). This discrepancy may be attributed to the fact that adult children, driven by their sense of filial piety, do not regard personal hedonic value as a priority that needs to be retained. Analysis results also underscore distinctions in the significance attributed to the three tourism values within the adult children's re-travel intention. Evidently, as is shown in Table 8, due to the differential impact of these distinct value types on attitudes (Table 7), the influence of positive interaction and role conflict on adult children's re-travel intention by shaping the three tourism values varies. In terms of the impacts of positive interaction, the relatively most pronounced influence is manifested through filial piety values ($\beta = 0.173$, $p < 0.001$), followed by the family connection value ($\beta = 0.137$, $p < 0.001$), whereas the personal hedonic value lacks significant impact. Similarly, in terms of the impacts of role conflict, the relatively most pronounced influence is also manifested through filial piety values ($\beta = -0.096$, $p < 0.001$), followed by the family connection value ($\beta = -0.078$, $p < 0.001$), and the personal hedonic value lacks significant impact too. This signifies that adult children have psychological preferences for specific aspects of "adult children–parents" family tourism values, reflecting a hierarchy of "parental benefit > familial benefit > self-benefit".

Therefore, another theoretical contribution of this study is to reveal the special psychology of adult children in the context of Confucian culture by analyzing the special form of "adult children–parents" family tourism.

Fourthly, self-efficacy moderates the impact paths between positive interactions, role conflicts, and family tourism values. Notably, this study shows that self-efficacy significantly impacts the formation process of family tourism values. Specifically, the higher the self-efficacy, the better the positive interaction's impact and the less impact the role conflict has. However, it is noteworthy that only the influence brought by a high self-efficacy on the formation of family connection value will impact adult children's re-travel intention. The theoretical contribution of this finding is that it clarifies the boundaries of the role of positive interaction and role conflict on family tourism values and provides a valuable addition and extension of the research on the impact of family tourists' interactive behaviors.

*5.2. Practical Implications*

The four important conclusions of this study have reference value for improving the effect of "adult children–parents" family tourism, increasing the adult children's intention to accompany their parents on family tourism, promoting the development of the family tourism market, and also providing some reference for the management activities of related tourism enterprises.

Firstly, tourism destination operators need to understand the forms of positive interaction between adult children and their parents in the process of family tourism and create a favorable environment for positive interaction between adult children and their parents by providing corresponding tourism projects and facilities, such as providing more activities suitable for adult children and parents to participate in and building a suitable background for taking family photos.

Second is the suppression of role conflict's negative effect on family tourism's value. This study finds that role conflict abates family tourism value. Tourism destination operators should accurately grasp the dual attributes of family tourism and improve the tourism experience of family tourists by continuously optimizing the tourism environment. For example, in response to differences in the choice of tourism programs within the family, destinations can reduce overall costs by opening more service windows, reducing queuing time, and providing program discount packages to satisfy the needs of all family members as much as possible.

Thirdly, this study confirms that filial piety value, family connection value, and attitude play mediating roles between positive interactions, role conflicts, and re-travel intention. Therefore, tourist destinations where family tourists visit more often need to evaluate the value of existing tourism products for tourists, strengthen the design and development of tourism products that meet the expectations of middle-aged and elderly people, and continuously optimize tourism products and services around "parents", so as to help adult children to better show their filial piety and prove themselves to their parents in family tourism. In addition, tourism destination operators should pay attention to the role of attitude, and take measures to enhance the positive attitude of adult children towards accompanying their parents on tourism, such as (1) utilizing diversified marketing channels to publicize the positive attitude of "adult children–parents" family tourism products through the new media channels for all ages, such as Tiktok, Weibo, and WeChat; (2) adopting multi-level marketing tools to directly improve adult children's attitudes toward accompanying their parents on family tourism through sensory marketing and affectionate marketing.

Fourthly, this study found that high self-efficacy will help improve the formation of family tourism value. This suggests that adult children should have prepared responses on how to resolve conflicts with parents that may occur during traveling, in addition to regular travel preparations during the preparation stage of family travel.

## 6. Conclusions

The purpose of this research is to investigate a particular psychological phenomenon that arises in adult Chinese children when they simultaneously assume the dual roles of children and tourists during "adult children–parents" family tourism. Therefore, based on the perspective of tourist-to-tourist interaction and role conflict theory, this study applied a "value–attitude–behavior" model to empirically analyze the relationship between the interactions of adult children and parents during the tourism and their re-travel intention. The results of the structural equation modeling (SEM) analysis indicate the following: (1) positive interactions have a positive impact on the formation of family tourism values; (2) role conflict has a negative impact on the formation of family tourism values; (3) positive interactions and role conflict both influence attitude through filial piety value and family connection value, which, in turn, affect adult children's re-travel intention; (4) self-efficacy moderates the relationship between positive interactions, role conflict, and family tourism values. However, it is important to be cautious when interpreting some causal relationships implied by the results of this study. Although the impact pathways within the model are based on widely accepted theoretical assumptions and are verified again in this research, it is noted that the path coefficients between role conflict and other variables are relatively small. This suggests that future research should consider additional dependent variables, particularly antecedents of role conflict.

## 7. Limitations

This study exhibits limitations due to research conditions and other factors.

Firstly, "family tourism values" is a multidimensional concept, but this research only involves three kinds: filial piety value, family connection value, and personal hedonic value. Further exploration of additional family tourism values remains vital for subsequent research endeavors.

Secondly, given that urbanization and economic development significantly impact family tourism frequency, this study selected regions with better conditions (Jiangsu, Zhejiang, and Shanghai) as research areas. Although researchers have performed some essential investigation in the preliminary research stage, considering the specificity of the research areas, whether the results can be applied to other areas in China still needs to be verified by subsequent research.

Thirdly, family tourism behavior is a complex behavioral mechanism, and studies have shown that the interactive behaviors of family travelers during tourism are also affected by a variety of demographic and family characteristics, as well as other factors, such as transportation mode and tourism forms. Limited to the time and other research conditions, this study did not look further into the differences caused by these characteristics.

## 8. Further Research

Future research will better the survey instrument through the incorporation of state-of-the-art advancements in the field of family tourism research, while additionally embracing a broader spectrum of potential variables, especially the individual family status (e.g., living alone; with others; with young children) [112] and tourism characteristics. Ongoing ameliorations to the foundational research model and the fine-tuning of investigative methodologies will be actively pursued.

**Author Contributions:** Conceptualization, A.H.; methodology, A.H. and H.L.; investigation, H.L.; formal analysis, A.H. and H.L.; writing—original draft preparation, A.H. and H.L.; writing—review and editing, J.P.; supervision, A.H.; funding acquisition, A.H. All authors have read and agreed to the published version of the manuscript.

**Funding:** This work was supported by the Major Project of National Social Science Fund of China (21&ZD119), National Natural Science Foundation of China (72074052) and Shanghai Planning of Philosophy and Social Science (2019BGL031).

**Institutional Review Board Statement:** Not applicable.

**Informed Consent Statement:** Informed consent was obtained from all subjects involved in the study.

**Data Availability Statement:** Not applicable.

**Conflicts of Interest:** The authors declare no conflict of interest. The funders had no role in the design of the study; in the collection, analyses, or interpretation of data; in the writing of the manuscript; or in the decision to publish the results.

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
