# Peer review of "A Tale of Two Identities: The Value, Attitude, and Behavior of Adult Children towards Family Tourism Experiences"

_sustainability, doi:10.3390/su151914364_

Round 1

Reviewer 1 Report

This is a review of the manuscript titled: “A tale of two identities: the value, attitude, and behavior of adult children towards family tourism experiences” that investigates the effects of tourism on adult children interaction with their parents by doing an empirical investigation obtaining the characteristics of the interactions, role conflict,  motivations and causal relationships. 

In general, I think you should have proposed a control group to compare the benefits of family tourism. Moreover, you need to describe more the conditions of the families and the possible differences with other forms of tourism in adult children. I am not sure if the families that can travel are not already privileged, and the results are only causative of a high economical class.  

Line 32. Include a definition of family tourism.

Please include the age ranges of adult children.  

What are the psychological effects of the commitment to parents and family? Does the concept have evolved and has lower the percentage of adult children that are committed to their family?

Line 65. If you refer to previous studies, include more than 1 reference (only reference 17).

Line 101. What unique value features are you referring to?

Line 122. Please include a definition of filial piety and examples of how it is beneficial.

A group of children compared with adult children could have given you a better approach to the information you have, 

I suggest including a 2 by 2 table with all the hypothesis and briefly referring to what do you want to test. 

Methods. As a supplementary material, please include the survey, even if you don't include it in the final version of the manuscript. Please indicate how do you approach the families for participating in the survey. Did they receive any incentive? How many were online and on paper?, How much time did they spend on the making of the survey.

Table 1. Please put headers of variables in bold. Also, it will help that the table would be divided into 3 column table, instead of 6 columns.

Please put monthly income in dollars to have a worldwide understanding of the economic conditions and if it refers to high, medium, or low class. 

Figures. They need more resolution and a bigger font. Also, they can be put one beside the other or centered on the page. 

Line 495. Would it be necessary to mention taking photos? What would be the significance of that activity?

Line 499. What are the differences of family travel without involvement of tourism agencies?, are there differences if the tourism is made by car, by bus, by train?, are the effects the same considering your variables?

Please mention tourism programs in the beginning

Reviewer 2 Report

This paper applied a "value-attitude-behavior" model to empirically analyze the relationship between the interactions of adult children and parents during the tourism and their re-travel intention. 

The three questions of the research are clearly stated in the introduction, as well as the rationale of the study. The hypotheses are also highlighted throughout section 2, based on an analysis of the existing literature and a conceptual framework is proposed and further on tested. This research employed a comprehensive array of data analytical methodologies, encompassing exploratory factor analysis (EFA), confirmatory factor analysis (CFA), structural equation modelling (SEM), and descriptive analysis.

The results are presented in detail in the fourth section and illustrated in 7 tables and 3 figures. The Discussion section reveals the theoretical contributions of this study (compared to previous research) as well as the managerial implications.

The limitations of the research are highlighted, s well as future research direction.

Based on these observations, I think that this paper may be published.

Reviewer 3 Report

Dear Authors,

I congratulate you for the idea of studying such a niche subject in tourism literature.

As for improving the quality of your manuscript, a few suggestions could be followed:

a. at first, please enrich the literature review that supports each of your hypotheses;

b. In regard with the Methodological approach, it should include:

-  participants and procedure;

- measures that you used;

- analysis strategy;

- results where you need to enclose descriptive statistics and hypotheses testing.

c. The pre-survey you are referring to is actually a pilot study? please motivate your answer.

d. You declare that you collected data in both formats: online and offline. You analysed data globally; how do you avoid bias in this regard? Please motivate in the light of previous literature.

e. Moreover: you applied questionnaires by engaging physical contact; additionally, you also used professional platforms for applying the same research instruments, but without any representativity and accuracy distinction. My concern is that you shouldn’t be able to analyse that data as a whole database. The representativity is clearly biased; you cannot ensure the fact that you don t have cases with the same respondent; moreover, the professional databases have respondents pools that might create a high risk of bias for you data.

f. As to solve this issue and not disengage the conformity of your methodological approach, please find previous literature methodological approaches (article/methodology books etc) that enhance your methodological proposal and validate your results.

g. Also, please motivate that such a case does not encounter a serious research ethics problem.

h. following the methodology, a Discussion section needs to be included, where the studied concepts and results need to be analyzed in regard with the statistical data.

i. moreover, the current study needs to have sections of Theoretical implications, Practical implications, Limitations of the study and future research (contributions of the study).

Hypothesizing that you can validate the above concerns, you should follow the next suggestions:

- all the results need to be validated with previous literature results and you should substantiate RMSEA CFI IFI and all the other indices scientifical accepted value ranges;

- discussion should include specific concepts dissemination in the light of previous literature results;

- please add a table where you display whether each hypothesis has been validated or not;

- please engage the limitations in regard to database issued discussed previously.

Best regards,

minor English editing errors

Reviewer 4 Report

I congratulate you on this study, which contains interesting and interesting results. I have indicated in the report the issues that I foresee to be corrected.

Round 2

Reviewer 1 Report

Dear authors,

Thank you for your efforts in improving the manuscript. I will suggest the acceptance of the article.

Best regards.

Reviewer 3 Report

Dear Authors,

Although it is visible the effort for improving your manuscript, a few errors need to be addressed:

- the hypotheses within the literature review need to be explained individually. What did you intend to say by H1a H1c?  you cannot have three hypotheses with the same content; please read Academic Methodology books and references. You might find some suggestions on https://www.bing.com/search?pglt=41&q=Sustainability+journal+methodology&cvid=417357e285ce44e6bc1d4a89620045eb&gs_lcrp=EgZjaHJvbWUyBggAEEUYOTIGCAEQRRhA0gEINzk1MGowajGoAgCwAgA&FORM=ANNTA1&PC=U531

The error repeats throughout the entire literature review.

- within section 3 – Data collection you motivate  your choice in regard to acquiring data; please provide literature examples with previous initiatives of this kind and that have  been published. In this regard, please read again point f from the previous review: f. As to solve this issue and not disengage the conformity of your methodological approach, please find previous literature methodological approaches (article/methodology books etc) that enhance your methodological proposal and validate your results

- section 4.3. – validity and reliability test for measures provides certain thresholds that you use in order to perform your analysis; please provide for each threshold literature references that validate your choices

You can follow examples such as

Sustainability | Free Full-Text | Leadership and Work Engagement Effectiveness within the Technology Era (mdpi.com)

COVID-19 and the workplace: Implications, issues, and insights for future research and action. (apa.org)

Please review the following sections – Conclusions/Limitations and further Research and enrich it.

Best regards,

Minor English spelling errors

Round 3

Reviewer 3 Report

Dear Authors,

I congratulate you for the final version of your manuscript. 

I wish you best of luck within your further research enquiries!

Best regards,

minor spelling issues detected